# The Effect of Indigo (*Indigofera tinctoria* L.) Waste on Growth Performance, Digestibility, Rumen Fermentation, Hematology and Immune Response in Growing Beef Cattle

**DOI:** 10.3390/ani13010084

**Published:** 2022-12-26

**Authors:** Nirawan Gunun, Chatchai Kaewpila, Waroon Khota, Sineenart Polyorach, Thachawech Kimprasit, Wasana Phlaetita, Anusorn Cherdthong, Metha Wanapat, Pongsatorn Gunun

**Affiliations:** 1Department of Animal Science, Faculty of Technology, Udon Thani Rajabhat University, Udon Thani 41000, Thailand; 2Department of Animal Science, Faculty of Natural Resources, Rajamangala University of Technology Isan, Sakon Nakhon Campus, Phangkhon, Sakon Nakhon 47160, Thailand; 3Department of Animal Production Technology and Fisheries, Faculty of Agricultural Technology, King Mongkut’s Institute of Technology Ladkrabang, Bangkok 10520, Thailand; 4Department of Plant Science, Faculty of Natural Resources, Rajamangala University of Technology Isan, Sakon Nakhon Campus, Phangkhon, Sakon Nakhon 47160, Thailand; 5Tropical Feed Resources Research and Development Center (TROFREC), Department of Animal Science, Faculty of Agriculture, Khon Kaen University, Khon Kaen 40002, Thailand

**Keywords:** indigo waste, feed intake, digestibility, average daily gain, volatile fatty acid, hematological indices, immune response, beef cattle

## Abstract

**Simple Summary:**

Indigo waste is a by-product of the processing of natural indigo dye. Indigo waste could be utilized as a protein source in ruminant rations, which would reduce the cost of feed. We evaluated the effects of the inclusion of indigo waste in concentrate diets on the feed utilization, rumen fermentation, hematology, immune function and growth performance of growing beef cattle. The present findings suggest that the inclusion of indigo waste at low levels in concentrate diets maintains feed intake, digestibility, rumen fermentation or growth performance in growing beef cattle without affecting hematology or immune function.

**Abstract:**

This experiment was conducted to assess the effect of indigo waste on the feed intake, digestibility, rumen fermentation, hematology, immune response and growth performance in growing beef cattle. Twenty crossbred beef cattle with an initial body weight (BW) of 145 ± 11 kg were fed four levels of indigo waste for 90 days in a trial. Additions of indigo waste at 0%, 10%, 20% and 30% in a concentrate diet using a completely randomized design (CRD). Cattle were fed concentrate at 1.8% BW, with rice straw fed ad libitum. The concentrate intake decreased linearly (*p* = 0.01) with the addition of indigo waste. The supplementation with indigo waste reduced dry matter (DM) and organic matter (OM) digestibility cubically (*p* = 0.03 and *p* = 0.02, respectively), while increasing neutral detergent fiber (NDF) digestibility cubically (*p* = 0.02). The final BW of beef cattle decreased linearly (*p* = 0.03) with the addition of indigo waste. The inclusion of indigo waste decreased the average daily gain (ADG) and gain-to-feed ratio (G:F) linearly (*p* < 0.01) from 0 to 90 days. The nutrient digestibility, ADG and G:F of beef cattle fed 10% indigo waste in the diet was similar when compared with the control (0% indigo waste). The ruminal pH, ammonia-nitrogen (NH_3_-N) and total volatile fatty acid (VFA) concentrations were similar among treatments (*p* > 0.05). The proportion of acetate increased linearly (*p* < 0.01) but propionate decreased linearly (*p* < 0.01), resulting in an increase in the acetate to propionate ratio (*p* < 0.01) when cattle were fed with indigo waste supplementation. Increasing indigo waste levels did not influence blood urea nitrogen (BUN) levels, hematological parameters or immune responses (IgA, IgM and IgG) (*p* > 0.05). In conclusion, the inclusion of indigo waste at 10% in a concentrate diet did not have a negative effect on feed intake, nutrient digestibility, rumen fermentation, hematology, immune function or growth performance in growing beef cattle.

## 1. Introduction

In tropical areas, most ruminants are fed poor-quality roughages, especially rice straw [1,2]. The inclusion of concentrate, which is high in protein and other nutrients, can greatly enhance the efficiency of ruminant production [3]. However, the high cost and competitiveness of feedstuffs, especially soybean meal, has led to the search for alternative feeds, especially agricultural and agro-industrial by-products [4,5]. The sustainability of using by-products as feed for ruminants is based on their nutritive value, the availability of nutrients, rumen fermentation patterns, production responses and feed cost compared to conventional diets [6]. Consequently, they recirculate these wastes into the food supply chain [7]. Researchers have suggested that their waste products can be used as a source of protein in concentrate mixtures for ruminants at a rate of 10–30% [8,9,10]. In addition, the proper utilization of agro-industrial by-products in ruminant nutrition and the finding of new, inexpensive feed resources that promote ecological sustainability in feedstuffs are necessary [11,12].

Indigo (*Indigofera tinctoria* L.) is a legume plant classified in the family Fabaceae and distributed in Africa, South Asia and South East Asia, especially in Thailand [13]. Indigo is used in a variety of industries, including coloring food, cosmetics and pharmaceuticals but most commonly in textile products [14]. The indigo leaf contains 30.5% CP, 2.4% EE, 19.0% crude fiber and 36.6% carbohydrate [15]. Moreover, the indigo stem contains 5.1% CP, 2.0% EE, 54.5% crude fiber and 30.0% carbohydrate [15], as well as some phytonutrients, including condensed tannins, saponins and flavonoids [16,17]. Muda et al. [17] reported that supplementation with indigo leaf extract reduced fecal egg count but had no effect on growth performance or hematology in sheep.

A report from Thailand’s Sakon Nakhon Province’s Community Development Department showed that there were 120 groups of producers and enterprises producing indigo dye in 2015. Natural indigo dye making with indigo plant biomass is claimed to produce the best indigo dye purity, according to the traditional method [18]. After the dye extraction process, the remaining indigo waste consists of the stem and leaves. Through the recycling of by-products indigo waste can be used as animal feed and low-cost feed opportunities [19]. Indigo waste is believed to be rich in protein and has the potential to be used as a protein source to replace soybean meal in ruminant diets and reduce the cost of feed. In addition, the indigo plant has successfully served as a source for antibacterial, antioxidant, anti-inflammatory and immunomodulatory effects [20,21,22,23]. We hypothesize that the use of indigo waste will maintain feed utilization and growth performance while improving rumen fermentation, hematology and immune function in beef cattle. Therefore, the objective of this study is to assess the effects of the inclusion of indigo waste in concentrate on feed intake, nutrient digestibility, rumen fermentation, growth performance, hematological and immunological responses in beef cattle.

## 2. Materials and Methods

### 2.1. Ethical Procedure

The Animals Ethical Committee of the Rajamangala University of Technology Isan approved the animal care and experimental procedures (approval number 24/2564).

### 2.2. Animals, Treatments and Experimental Design

The study was conducted on the beef cattle farm of the Faculty of Natural Resources at Rajamangala University of Technology Isan, Sakon Nakhon Campus in Phangkhon, Sakon Nakhon, Thailand. After the dye extraction process, fresh indigo waste (leaf and stem) was collected from the indigo cloth community enterprise group, Baan Nohnrua, Pannanikom, Sakon Nakhon, Thailand. They were sun-dried for four days, then ground before being added to the concentrate.

Twenty male crossbred (Brahman × Thai native) beef cattle with an average weight of 145 ± 11 kg were raised for a 90-day experiment. Each cattle was contained in a separate pen with access to fresh water. The cattle were fed concentrate at a rate of 1.8% of their body weight (BW), along with rice straw ad libitum, in two equal feedings at 08:00 h and 16:00 h. This study was conducted using a completely randomized design (CRD) to compare the indigo waste included in the concentrate at 0%, 10%, 20% and 30% on a DM basis (Table 1).

### 2.3. Feed Costs Analysis

The feed costs of the diets containing indigo waste were calculated using an input budgeting procedure according to Serrapica et al. [24]. However, the average costs of feedstuffs at the local suppliers’ gate were used in our calculation. The feed costs were adjusted based on the actual DM content and converted from baht to USD using 0.0286 currency. The feed costs (USD/kg DM) were cassava chip 0.32, rice bran 0.23, soybean meal 0.76, dried brewers’ grains 0.41, indigo waste 0.06, molasses 0.29, mineral and vitamin mixture 1.51, urea 0.86, salt 0.29 and sulfur 0.91.

### 2.4. Data Collection and Sampling Procedures

Average daily gain (ADG) was estimated by weighing cattle at the beginning BW, 30 days, 60 days and the final BW at 90 days. Each morning, the offered and refused feed were recorded and taken for chemical analysis. Fecal samples were collected 56–60 days into the trial to conduct a digestibility test. Rectal sampling was used to obtain fresh feces (about 500 g). Each cattle’s daily fresh fecal samples were pooled and chilled at 4 °C. Samples of feeds, refusals and feces were dried at 60 °C and ground (1-millimeter screen using the Cyclotech Mill; Tecator, Hoganas, Sweden). The contents of ash, ether extract (EE), crude protein (CP) [25], NDF and acid detergent fiber (ADF) [25,26] were determined. The modified vanillin-HCl procedure based on Burns [27] was used to measure the indigo waste’s condensed tannin (CT) concentration. Methanol extraction was used to evaluate the crude saponins, as described by Kwon et al. [28] and modified by Poungchompu et al. [29]. The gross energy (GE) of the feeds was assessed by bomb calorimetry using an Oxygen bomb calorimeter (Parr Instrument Company, Moline, IL, USA), and acid-insoluble ash (AIA) was determined in the samples. AIA was created to assess the digestibility of nutrients [30].

On the 60th day of the experiment, 4 h after feeding, 200 mL of rumen fluid was taken with a stomach tube and a vacuum pump. The first 100 mL of the ruminal samples was thrown away to avoid contaminating them with saliva. The samples were then passed through four layers of cheesecloth and tested right away with a portable pH meter. Ruminal fluid samples were centrifuged at 16,000× *g* for 15 min at 4 °C, and the supernatant was kept at −20 °C. The ruminal samples were thawed and utilized to analyze NH_3_-N (Kjeltech Auto 1030 Analyzer, Tecator, Hoganas, Sweden) [31] and VFA (GC 8890; Agilent Technologies Ltd., Santa Clara County, CA, USA) [32].

Blood samples were taken at the same time as rumen fluid samples. Here, 10 mL of fresh blood was taken from the jugular vein of each animal. Blood urea nitrogen (BUN) was measured by Crocker’s method [33]. A hematological analyzer (BCC-3000B; DIRUI, Gungoren/Istanbul, Turkey) was used to measure red blood cells (RBCs), hemoglobin, hematocrit, mean corpuscular volume (MCV), mean corpuscular hemoglobin (MCH), white blood cells (WBCs), neutrophils, lymphocytes, monocytes, eosinophils and platelet count. The concentrations of immunoglobulins (IgA, IgG and IgM) were measured using the nephelometric method (Mispa-i3, Agappe Diagnostics Ltd., Ernakulam, Kerala, India).

### 2.5. Statistical Analysis

Using a completely randomized design, the general linear model (GLM) in SAS software was utilized to evaluate the data for variances [34]. Orthogonal polynomial contrasts (linear, quadratic and cubic) were used to compare the treatment trends statistically. To determine if an effect was significant, a *p* < 0.05 significance level was used.

## 3. Results

### 3.1. Feed Cost Analysis and Chemical Composition of Diets

The feed costs of the diets prepared without and with increasing indigo waste levels are presented in Table 2. Feed costs ranged from 29.25 to 39.59 USD/100 kg DM. The safe costs according to the replacements of indigo waste with soybean meal and dried brewer’s grains were −3.63, −7.08 and −10.35 USD/100 kg DM, respectively, for 10%, 20% and 30%. The indigo waste consists of 19.8% of CP, 46.6% of NDF, 32.4% of ADF, 5.4% of CT, 13.1% of saponins and 3487.5 kcal/kg DM of GE (Table 3). The local feed resources were used to create the concentrate diets, which had a 14.3–14.6% CP content. The NDF, ADF and GE content were increased according to the inclusion of indigo waste in the concentrate.

### 3.2. Feed Intake and Digestibility

The increasing levels of indigo waste decreased linearly (*p* = 0.01) the concentrate intake from 0 to 90 days, but it did not affect the roughage intake (*p* > 0.05) (Table 4). The inclusion of indigo waste decreased total intake linearly (*p* = 0.02) from 61 to 90 days. The digestibility of CP and ADF was similar among the groups (*p* > 0.05), whereas DM and OM digestibility decreased cubically (*p* = 0.03 and *p* = 0.02, respectively) with increasing levels of indigo waste (Table 5). Furthermore, the digestibility of DM and OM were lowest in cattle fed with the inclusion of indigo waste at 30% in concentrate. The NDF digestibility increased cubically with the addition of indigo waste (*p* = 0.02).

### 3.3. Performance

The BW in the cattle trial was similar among treatments (*p* > 0.05) on days 0, 30 and 60, whereas the BW on days 90 decreased linearly with the addition of indigo waste (*p* = 0.03) (Table 6). The ADG and G:F decreased linearly (*p* < 0.01) with increasing levels of indigo waste from 0 to 90 days. In addition, ADG and G:F were lower with the addition of indigo waste at 20–30% in concentrate diets.

### 3.4. Rumen Fermentation

The use of indigo waste in concentrate feed for cattle also had no effect on ruminal pH, NH_3_-N or total VFA (*p* > 0.05) (Table 7). The proportions of propionate (C3), valerate (C5) and iso-valerate (i-C5) decreased linearly (*p* < 0.01, *p* < 0.01 and *p* = 0.01), while acetate (C2) and C2:C3 increased linearly (*p* < 0.01) by including levels of indigo waste.

### 3.5. Blood Urea Nitrogen and Hematological Parameters

The increasing levels of indigo waste did not affect the concentrations of BUN, RBCs, hemoglobin, hematocrit, MCV, MCH, WBCs, neutrophils, lymphocytes, monocytes, eosinophils and platelet count (*p* > 0.05) (Table 8).

### 3.6. Immune Response

The use of indigo waste in concentrate feed for beef cattle had no effect on IgA, IgM and IgG (*p* > 0.05) (Table 9).

## 4. Discussion

The CP content of the indigo waste was 19.8%. However, Bhatta et al. [16] indicated that the CP of indigo leaf was 26.0% DM. Because indigo waste is composed of leaves and stems, it has a lower CP than indigo leaf. The inclusion of indigo waste increases the fiber and GE content in the concentrate. The indigo waste contained NDF, ADF and GE at 46.6%, 32.4% DM and 3487.5 kcal/kg DM, respectively. The treatment chemical composition indicates that adding indigo waste increased the fiber concentration and gross energy.

It is generally accepted that the NDF concentrations of the diet, which limit the DM digestibility of the diet, are a major factor limiting the voluntary DM intake of animals [35]. The inclusion of indigo waste in concentrate diets decreased concentrate intake, total intake and DM digestibility in the present study. Indigo waste (stem and leaf) has a high fiber content. Increased fiber content in concentrate diets with the addition of indigo waste results in reduced feed intake and the digestibility of DM and OM with increasing levels of indigo waste. Moreover, indigo waste has a high GE content, and GE increases when indigo waste levels are increased in concentrate diets. However, DM and OM digestibility declined. This could be due to the high-fiber energy of the by-product with reduced digestible energy when the addition of indigo waste, especially at 30% in concentrate, results in a decrease in beef cattle’s DM and OM digestibility. The greater NDF digestibility of the diet was mainly due to enhanced hemicellulose digestion [36]. In the current study, increasing levels of indigo waste by 20% in concentrate diets improved digestibility. Similarly, Kongphitee et al. [37] reported that NDF digestibility increased with increasing levels of by-product in the diets of beef cattle. Lyu et al. [38] reported that the inclusion of by-products in diets enhanced the digestibility of NDF in dairy cows. However, ADF digestibility was not affected by indigo waste supplementation. This is plausible because a higher fraction of the hemicellulose is present in indigo waste in concentrated diets, and therefore, NDF digestibility is increased but cellulose digestion is not affected.

ADG and feed efficiency are essential components of growing beef cattle production efficiency. ADG is an essential component of growing beef cattle production efficiency [39]. There was a linear decrease in ADG and G:F with an increasing level of indigo waste in growing beef cattle from 0 to 90 days. The ADG as well as the G:F ratio appear to be lower when cattle were fed diets containing 20–30% indigo waste from 0 to 90 days. Kanjanapruthipong et al. [40] reported that increasing NDF content in the diets decreased the digestibility of nutrients and ADG in dairy cattle. These results may be due to the high NDF content and also the lower DM intake, nutrient availability, and VFA, particularly propionate, when indigo waste was gradually increased in concentrate diets, resulting in decreased growth performance in growing beef cattle. The ADG required to obtain the target body weight is based on the body weight at the beginning of the trial and feed efficiency. The addition of indigo waste reduces the ADG and G:F, thereby decreasing the final BW (90 days of a trial). These results suggest that the use of indigo waste at 10% in concentrate diets is suitable for growth performance in growing beef cattle.

Ruminal pH is among the major fermentation factors that directly affect microbial ecology and, thereby, ruminal fermentation [41]. In our experiment, the rumen pH range for all diets was 6.8 to 6.9, and the optimum range for microbial activity in the rumen was 6.5–7.0 [4,42]. The ruminal pH was similar among treatments, which indicates that the inclusion of indigo waste did not change rumen ecology or fermentation in tropical beef cattle. The main nitrogen source for protein synthesis in the rumen is NH_3_-N [8]. In the present investigation, indigo waste levels had no effect on the ruminal NH_3_-N concentration, indicating that indigo waste seemed to have no effect on protein degradation by microorganisms in the rumen. The ruminal NH_3_-N concentrations ranged from 16.8 to 21.5 mg/dL, which is closer to the optimum range (15 to 30 mg/dL) [1,43].

The pattern of rumen fermentation was changed by the different diets fed to ruminants. As VFA production serves as an energy source for growth performance in ruminants, it is crucial to understand their metabolism [44]. Acetate, propionate and butyrate are the main VFAs produced in the rumen, and their concentrations vary depending on the feed ingredient, feed intake, digestibility, rumen ecology and the rate of passage [45]. When indigo waste was added to the diets, the rumen VFA profile changed, with an increase in acetate and a decline in propionate; this also caused the C2:C3 ratio to increase. The association between the C2:C3 ratio and feed has been explained by the metabolic properties of fiber- and starch-degrading bacteria [46]. Most structural carbohydrate fermentation, which leads to the production of acetate, is caused by cellulolytic bacteria. The major cellulolytic bacteria are thought to be *Ruminococcus albus*, *R. flavefaciens* and *Fibrobacter succinogenes*, which produce more acetate in the rumen [47]. Several different types of bacteria, including those in the family *Propionibacteriaceae*, produce propionate as an end product in the rumen [48]. A high concentration of starch in the diet is more likely to ferment into propionate production in the rumen, making it advantageous for the production of glucose, which helps the meat animal. [49]. The addition of a high-fiber by-product feed increased acetate levels while reducing propionate in the rumen [50]. Wanapat et al. [46] found that adding high amounts of structural carbohydrates to the diet increased the proportions of acetate in the rumen, which caused the C2:C3 ratio to be higher. This means that indigo waste diets had more fermentable structural carbohydrates, such as hemicellulose, which is thought to increase acetate production and decrease propionate production.

The BUN concentration is often used to assess protein supplies and metabolic concerns related to animal diseases [51]. The addition of indigo waste to concentrate had no effect on BUN, which ranged from 8.2 to 12.0 mg/dL, which was within the usual range of 8 to 14 mg/dL in tropical beef cattle [52,53]. Cattle health and nutrition and the cause of an abnormality or malfunction in cattle are frequently tested using hematological analysis [42,54]. The inclusion of indigo waste did not influence all hematological indicators. Similarly, Muda et al. [17] reported that the hematology of sheep was not affected by indigo leaf extract supplementation. These results indicate that the addition of indigo waste as a feed had no negative effects on the health status of tropical beef cattle. Growing beef cattle had normal concentrations of RBCs, hemoglobin, hematocrit, WBCs, neutrophils, lymphocytes and eosinophils when compared to our previous study [42,51,55]. In addition, previous reports have demonstrated that concentrations of MCV, MCH [51,56], monocytes [57] and platelet count [57,58] in ruminant blood are within the accepted range.

Phytonutrients from tropical plants, which have been thought of as possible additions to animal feed, may affect immune and inflammatory responses. There was a high amount of phytonutrients such as total phenolics, total tannins, saponins and flavonoids in the indigo [22]. Indigo leaf extract has numerous pharmacological effects, including anti-inflammatory, antioxidant, antibacterial, antiviral and other activities [59]. In addition, this plant improved the immune response and demonstrated its immunostimulating efficacy in vitro and in rats [20,21]. In the current study, the addition of indigo waste had no influence on IgA, IgM or IgG immune responses. These results suggest that some phytonutrients may be water-soluble during the dye extraction procedure or that indigo waste contains crude leaf and stem, resulting in a weak effect on the immune response in cattle.

## 5. Conclusions

The inclusion of indigo waste in the concentrate did not affect the NH_3_-N concentrations, hematological parameters or immune response. However, the addition of indigo waste had an effect on the feed intake, digestibility of DM and OM and growth performance. Furthermore, incorporating indigo waste into the diets reduced propionate while increasing the acetate proportion. The addition of 10% indigo waste to the concentrate showed that it could be used as a source of protein and sustain growing beef cattle feed intake, nutrient digestibility, rumen fermentation and growth performance. In order to assess the effects of indigo waste on carcass characteristics and meat quality in beef cattle, more research needs to be conducted.

## Figures and Tables

**Table 1 animals-13-00084-t001:** Ingredients of the diet used in the experiment.

Item	Level of Indigo Waste (%DM)			
	0	10	20	30
Ingredient, kg dry matter (DM)				
Cassava chip	45.0	45.0	45.0	45.0
Rice bran	19.0	14.0	9.0	5.0
Soybean meal	14.0	11.0	8.5	7.0
Dried brewers’ grains	17.5	15.5	13.0	8.5
Indigo waste	0.0	10.0	20.0	30.0
Molasses	2.0	2.0	2.0	2.0
Mineral and vitamin mixture	1.0	1.0	1.0	1.0
Urea	0.5	0.5	0.5	0.5
Salt	0.5	0.5	0.5	0.5
Sulfur	0.5	0.5	0.5	0.5

**Table 2 animals-13-00084-t002:** Feed costs (USD/100 kg DM) of the experimental diets prepared with indigo waste.

Item	Level of Indigo Waste (%DM)			
	0	10	20	30
Cassava chip	14.29	14.29	14.29	14.29
Rice bran	4.41	3.25	2.09	1.16
Soybean meal	10.68	8.39	6.48	5.34
Dried brewers grains	7.10	6.29	5.28	3.45
Indigo waste	0.00	0.63	1.26	1.90
Urea	0.43	0.43	0.43	0.43
Molasses	0.57	0.57	0.57	0.57
Mineral and vitamin mixture	1.51	1.51	1.51	1.51
Salt	0.14	0.14	0.14	0.14
Sulfur	0.46	0.46	0.46	0.46
Total feeding costs	39.59	35.96	32.51	29.25
Safe costs (vs 0% indigo waste)	0.00	−3.63	−7.08	−10.35

**Table 3 animals-13-00084-t003:** Chemical composition of concentrate, rice straw and indigo waste.

Item	Level of Indigo Waste (%DM)				Rice Straw	Indigo Waste
	0	10	20	30		
Chemical composition						
Dry matter, %	87.9	87.2	87.7	87.3	91.0	89.7
Organic matter, %DM	90.1	90.4	92.2	92.3	87.8	90.5
Crude protein, %DM	14.6	14.3	14.4	14.5	4.9	19.8
Neutral detergent fiber, %DM	43.1	46.0	52.1	59.5	73.7	46.6
Acid detergent fiber, %DM	27.8	28.1	29.8	30.5	51.9	32.4
Ash, %DM	9.9	9.6	7.8	7.7	12.2	9.5
Gross energy, kcal/kg DM	2861.2	3048.7	3214.8	3580.8	2797.9	3487.5
Condensed tannins, %DM	-	-	-	-	-	5.4
Crude saponins, %DM	-	-	-	-	-	13.1

**Table 4 animals-13-00084-t004:** Effect of indigo waste on feed intake in growing beef cattle.

Item	Level of Indigo Waste (%DM)				SEM	Contrast		
	0	10	20	30		Linear	Quadratic	Cubic
Dry matter intake, kg/d								
Concentrate								
0 to 30 d	2.8	2.5	2.4	2.4	0.19	0.11	0.61	0.93
31 to 60 d	3.4	3.1	2.7	2.6	0.22	0.01	0.69	0.59
61 to 90 d	3.8	3.5	3.1	2.8	0.26	<0.01	0.94	0.98
0 to 90 d	3.3	3.0	2.7	2.6	0.21	0.01	0.79	0.87
Roughage								
0 to 30 d	1.9	1.7	1.8	1.8	0.21	0.69	0.52	0.73
31 to 60 d	2.2	1.9	1.9	1.9	0.20	0.38	0.32	0.75
61 to 90 d	2.2	2.0	2.0	2.0	0.21	0.42	0.65	0.74
0 to 90 d	2.1	1.9	1.9	1.9	0.19	0.47	0.47	0.73
Total intake								
0 to 30 d	4.7	4.2	4.2	4.2	0.36	0.27	0.52	0.81
31 to 60 d	5.6	5.0	4.6	4.6	0.40	0.07	0.47	0.88
61 to 90 d	6.1	5.5	5.2	4.7	0.39	0.02	0.84	0.85
0 to 90 d	5.5	4.9	4.7	4.5	0.37	0.07	0.60	0.93

**Table 5 animals-13-00084-t005:** Effect of indigo waste on nutrient digestibility in growing beef cattle.

Item	Level of Indigo Waste (%DM)				SEM	Contrast		
	0	10	20	30		Linear	Quadratic	Cubic
Digestibility, %								
Dry matter	54.9	55.3	60.2	51.3	1.75	0.48	0.02	0.03
Organic matter	58.8	58.7	64.1	55.2	1.74	0.50	0.02	0.02
Crude protein	49.3	48.3	49.3	47.3	2.85	0.74	0.86	0.70
Neutral detergent fiber	49.1	50.8	60.8	56.1	1.90	<0.01	0.13	0.02
Acid detergent fiber	46.9	43.2	47.2	44.2	1.75	0.63	0.86	0.10

**Table 6 animals-13-00084-t006:** Effect of indigo waste on growth performance in growing beef cattle.

Item	Level of Indigo Waste (%DM)				SEM	Contrast		
	0	10	20	30		Linear	Quadratic	Cubic
Body weight, kg								
Initial	156.4	138.4	140.8	143.6	12.33	0.52	0.41	0.72
30 d	185.2	167.2	158.4	159.6	13.05	0.16	0.47	0.98
60 d	213.6	191.8	179.2	176.0	14.52	0.07	0.53	0.99
Final	238.4	214.2	197.6	190.4	15.55	0.03	0.59	0.98
ADG, kg/d								
0 to 30 d	0.96	0.96	0.58	0.52	0.09	<0.01	0.76	0.13
31 to 60 d	0.94	0.80	0.70	0.54	0.07	<0.01	0.89	0.76
61 to 90 d	0.82	0.76	0.62	0.48	0.09	0.01	0.67	0.85
0 to 90 d	0.91	0.84	0.63	0.51	0.08	<0.01	0.45	0.82
G:F								
0 to 30 d	0.20	0.22	0.15	0.13	0.03	0.02	0.46	0.14
31 to 60 d	0.18	0.16	0.15	0.12	0.01	0.02	0.64	0.93
61 to 90 d	0.14	0.14	0.12	0.10	0.02	0.04	0.51	0.86
0 to 90 d	0.17	0.18	0.14	0.12	0.06	<0.01	0.41	0.33

**Table 7 animals-13-00084-t007:** Effect of indigo waste on rumen fermentation in growing beef cattle.

Item	Level of Indigo Waste (%DM)				SEM	Contrast		
	0	10	20	30		Linear	Quadratic	Cubic
pH	6.8	6.9	6.9	6.9	0.07	0.38	0.23	0.78
NH_3_-N, mg/dL	19.6	21.5	20.6	16.8	1.81	0.26	0.14	1.00
Total VFA, mmol/d	54.7	54.6	58.0	59.8	2.79	0.15	0.73	0.68
VFA, mol/100 mol								
Acetate (C2)	58.1	60.8	61.3	62.7	0.70	<0.01	0.37	0.32
Propionate (C3)	24.2	20.2	20.1	18.6	1.10	<0.01	0.26	0.29
Butyrate (C4)	13.8	15.4	15.1	15.5	0.77	0.19	0.45	0.42
Iso-butyrate (i-C4)	0.9	0.9	1.0	0.8	0.09	0.63	0.29	0.46
Valerate (C5)	1.5	1.4	1.3	1.3	0.05	<0.01	0.13	0.72
Iso-valerate (i-C5)	1.4	1.3	1.2	1.0	0.31	0.01	0.91	0.74
C2:C3	2.4	3.0	3.1	3.5	0.40	<0.01	0.58	0.27

**Table 8 animals-13-00084-t008:** Effect of indigo waste on BUN and hematology in growing beef cattle.

Item	Level of Indigo Waste (%DM)				SEM	Contrast		
	0	10	20	30		Linear	Quadratic	Cubic
BUN, mg/dL	8.2	12.0	10.4	9.4	1.64	0.78	0.16	0.42
Red blood cells, 10^12^/L	4.9	4.8	4.9	5.1	0.40	0.73	0.71	0.99
Hemoglobin, g/dL	7.3	7.2	7.5	7.6	0.66	0.73	0.92	0.87
Hematocrit, %	22.8	21.8	22.6	22.8	1.97	0.92	0.76	0.78
MCV, 10^6^/fL	46.2	45.8	46.2	45.4	0.43	0.32	0.65	0.32
MCH, pg	21.8	20.6	19.8	20.6	1.32	0.46	0.46	0.84
White blood cells, 10^9^/L	13.9	14.2	10.7	12.9	1.79	0.42	0.59	0.25
Neutrophils, %	30.8	34.6	25.2	30.4	3.89	0.55	0.85	0.13
Lymphocytes, %	67.8	65.0	74.2	68.8	4.01	0.50	0.75	0.15
Monocytes, %	0	0	0	0	NA	NA	NA	NA
Eosinophils, %	1.4	0.4	0.6	0.8	0.57	0.54	0.31	0.64
Platelet count, 10^9^/L	214.0	266.0	274.3	219.0	33.98	0.88	0.17	0.91

MCV—mean corpuscular volume; MCH—mean corpuscular hemoglobin.

**Table 9 animals-13-00084-t009:** Effect of indigo waste on immune response in growing beef cattle.

Item	Level of Indigo Waste (%DM)				SEM	Contrast		
	0	10	20	30		Linear	Quadratic	Cubic
IgA, mg/dL	85.6	90.2	89.8	94.0	3.03	0.09	0.94	0.49
IgM, mg/dL	49.2	48.2	48.4	40.8	3.93	0.17	0.41	0.61
IgG, mg/dL	429.2	423.4	396.6	414.2	18.4	0.39	0.53	0.44

## Data Availability

Not applicable.

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
