# Peer review of "The Effect of Indigo (Indigofera tinctoria L.) Waste on Growth Performance, Digestibility, Rumen Fermentation, Hematology and Immune Response in Growing Beef Cattle"

_animals, 2022, doi:10.3390/ani13010084_

Round 1

Reviewer 1 Report

Dear authors,

I reviewed the manuscript identified as animals-2119406 (The Effect of Indigo (Indigofera tinctoria L.) Waste on Growth Performance, Digestibility, Rumen Fermentation, Hematology and Immune Response in Growing Beef Cattle). In my opinion, the manuscript is quite interesting and presents an agro-industrial by-product as a new potential feedstuff for ruminants feeding. The theme is relevant, particularly from a circular economy perspective, and the manuscript was produced based on solid science. However, I have some comments.

The studied by-product is not known to a wide audience. A graphic abstract would be VERY important for this manuscript. I recommend one figure (panel) depicting the indigo plant, the parts of the plant used in the industrial extraction, the commercial product, and the by-product being studied. Thanks.

In the introduction (L 64), the authors say that industrial indigo waste poses a hazard to the environment. I find it helpful that the authors support their claim. Among other things, I am surprised that plant biomasses potentially harmful to the environment can be safely used in animal feed and vice versa. In my opinion, the authors could more fruitfully refer to the concept of circular economy, emphasizing the opportunity to reuse in feeding animals’ precious biomass that would otherwise be disposed of. Thanks.

The acronym CDR (L 91) is not explained. As per journal's standard, acronymous could be explained at first mention. I invite the authors to check through the text. As a further suggestion, I would avoid starting the sentences with acronyms (i.e., L 96 among others).

Regarding the sampling procedures, the authors state that animals are housed in individual pens for 5 days to evaluate diets digestibility (L 98). However, I cannot see any differences with respect to the general management of the animals throughout the trial (see L 88-89). The authors could better explain what differences in animal handling there are between the two parts of the experiment if any. Thanks.

Table 2 shows the estimated price per kg of the experimental diets (I suppose this is the case). Rather than prices, I think the authors meant costs. Anyhow, no description of the calculation procedures is reported. Therefore, the authors should indicate the price of the raw materials used in the experimental diets’ formulation, which markets they refer to, and at which point of the supply chain the raw materials have been quoted (at the farm gate, for example, or other). Moreover, since the indigo waste replaces soybean meal along with other ingredients, the authors could usefully add an economic analysis. This evaluation, in my opinion, could give added value to the manuscript. In this regard, the authors refer to doi.org/10.3390/ani11082353, where are summarized the feeding cost calculation procedure in a stepwise replacement study. Thanks.

Since the journal does not impose a separate discussion of each result, the authors could reword the discussions in one paragraph, in my opinion.

Author Response

The studied by-product is not known to a wide audience. A graphic abstract would be VERY important for this manuscript. I recommend one figure (panel) depicting the indigo plant, the parts of the plant used in the industrial extraction, the commercial product, and the by-product being studied. Thanks.

- L48: Thank you for your suggestion. Already added graphical abstract, please see in the text.

In the introduction (L 64), the authors say that industrial indigo waste poses a hazard to the environment. I find it helpful that the authors support their claim. Among other things, I am surprised that plant biomasses potentially harmful to the environment can be safely used in animal feed and vice versa. In my opinion, the authors could more fruitfully refer to the concept of circular economy, emphasizing the opportunity to reuse in feeding animals’ precious biomass that would otherwise be disposed of. Thanks.

- L80-82: Already changed to “Through the recycling of by-products that can be used as animal feed and low-cost feed opportunities [19].”, please see in text.

The acronym CDR (L 91) is not explained. As per journal's standard, acronymous could be explained at first mention. I invite the authors to check through the text. As a further suggestion, I would avoid starting the sentences with acronyms (i.e., L 96 among others).

- Already changed to “….….completely randomized design (CRD) to compare the indigo waste included in the concentrate at 0%, 10%, 20%, and 30% on a DM basis (L106)”, and “Average daily gain (ADG) was estimated by weighing cattle at the beginning BW (L120)”, please see in text.

Regarding the sampling procedures, the authors state that animals are housed in individual pens for 5 days to evaluate diets digestibility (L 98). However, I cannot see any differences with respect to the general management of the animals throughout the trial (see L 88-89). The authors could better explain what differences in animal handling there are between the two parts of the experiment if any. Thanks.

- When the nutrient digestibility and fecal sampling evaluations at 56-60 days of the trials (L122-123) are the same as the trial management (L103-105).

- L122-123: Already changed to “Fecal samples were collected 56-60 days into the trial to conduct a digestibility test.”, please see in text.

Table 2 shows the estimated price per kg of the experimental diets (I suppose this is the case). Rather than prices, I think the authors meant costs. Anyhow, no description of the calculation procedures is reported. Therefore, the authors should indicate the price of the raw materials used in the experimental diets’ formulation, which markets they refer to, and at which point of the supply chain the raw materials have been quoted (at the farm gate, for example, or other). Moreover, since the indigo waste replaces soybean meal along with other ingredients, the authors could usefully add an economic analysis. This evaluation, in my opinion, could give added value to the manuscript. In this regard, the authors refer to doi.org/10.3390/ani11082353, where are summarized the feeding cost calculation procedure in a stepwise replacement study. Thanks.

- The feed costs of the diets prepared without and with increasing indigo waste levels were presented in Table 2.

- L111-117: Already added to “2.3 Feed costs analysis

The feed costs of the diets containing indigo waste were calculated using an input budgeting procedure according to Serrapica et al. [24]. However, the average costs of feedstuffs at the local suppliers’ gate were used in our calculation. The feed costs were adjusted based on the actual DM content and converted from Bath to USD using 0.0286 currency. The feed costs (USD/kg DM) were cassava chip 0.32, rice bran 0.23, soybean meal 0.76, dried brewers’ grains 0.41, indigo waste 0.06, molasses 0.29, mineral and vitamin mixture 1.51, urea 0.86, salt 0.29, and sulfur 0.91.”, please see in text.

- L157-161: Already added to “The feed costs of the diets prepared without and with increasing indigo waste levels were presented in Table 2. Feed costs ranged from 29.25 to 39.59 USD/100 kg DM. The safe costs according to the replacements of indigo waste with soybean meal and dried brewer's grains were -3.63, -7.08, and -10.35 USD/100 kg DM, respectively, for 10%, 20%, and 30%.”, please see in text.

Since the journal does not impose a separate discussion of each result, the authors could reword the discussions in one paragraph, in my opinion.

- Already changed in the discussions, please see in text.

Reviewer 2 Report

The manuscript contains valuable data. Results were properly reported, and the findings have been accurately discussed and compared with other published papers. For further improvement of the manuscript, it requires some modification.

P1,L22: Change “the diet of ruminants” to be “ruminants ration”

P1,L33: Change “unrestricted” to be “ad-libitum”

P2,L51: The introduction needs to be entirely re-written. It is very vague, and does not give the reader the necessary context to understand why you used the treatments you did, and why you made the measurements that you did. 

P7,L183-184: Change “Findings indicate that adding indigo waste increased the concentration of fiber and energy in the concentrate.” to be “Treatments chemical composition indicate that adding indigo waste increased the fiber concentration and gross energy”

P8,L245-247: The discussion needs to be much deeper.  The discussion is generally a re-statement of the results, a list of similar studies that agree or disagree with your results (without discussing why those papers support your results or contradict your results).

P9,L300: Some references are too old, Replace with new reference (2015-2022).

For using of wastes as animal feed, you can use this references:

Palangi, V., Kaya, A., Kaya, A., & Giannenas, I. (2022). Ecofriendly Usability of Mushroom Cultivation Substrate as a Ruminant Feed: Anaerobic Digestion Using Gas Production Techniques. Animals, 12(12), 1583.

Besharati, M., Palangi, V., Moaddab, M., Nemati, Z., Pliego, A. B., & Salem, A. Z. (2021). Influence of cinnamon essential oil and monensin on ruminal biogas kinetics of waste pomegranate seeds as a biofriendly agriculture environment. Waste and Biomass Valorization, 12(5), 2333-2342.

Author Response

The manuscript contains valuable data. Results were properly reported, and the findings have been accurately discussed and compared with other published papers. For further improvement of the manuscript, it requires some modification.

P1,L22: Change “the diet of ruminants” to be “ruminants ration”

- L22: Already changed, please see in text.

P1,L33: Change “unrestricted” to be “ad-libitum”

- L33: Already changed, please see in text.

P2,L51: The introduction needs to be entirely re-written. It is very vague, and does not give the reader the necessary context to understand why you used the treatments you did, and why you made the measurements that you did.

- L53-90: Already re-written for the introduction, please see in text.

P7,L183-184: Change “Findings indicate that adding indigo waste increased the concentration of fiber and energy in the concentrate.” to be “Treatments chemical composition indicate that adding indigo waste increased the fiber concentration and gross energy”

- L214-215: Already changed, please see in text.

P8,L245-247: The discussion needs to be much deeper.  The discussion is generally a re-statement of the results, a list of similar studies that agree or disagree with your results (without discussing why those papers support your results or contradict your results).

- L270-279: Already changed to “Several different types of bacteria, including those in the family Propionibacteriaceae, pro-duce propionate as an end product in the rumen [50]. A high concentration of starch in the diet is more likely to ferment into propionate production in the rumen, making it advanta-geous for the production of glucose, which helps the meat animal. [51]. The addition of a high-fiber by-product feed increased acetate levels while reducing propionate in the rumen [52]. Wanapat et al. [48] found that adding high amounts of structural carbohydrates to the diet increased the proportions of acetate in the rumen, which caused the C2:C3 ratio to be higher. This means that indigo waste diets had more fermentable structural carbohydrates, like hemicellulose, which is thought to increase acetate production and decrease propionate production.”, please see in text.

P9,L300: Some references are too old, Replace with new reference (2015-2022).

For using of wastes as animal feed, you can use this references:

Palangi, V., Kaya, A., Kaya, A., & Giannenas, I. (2022). Ecofriendly Usability of Mushroom Cultivation Substrate as a Ruminant Feed: Anaerobic Digestion Using Gas Production Techniques. Animals, 12(12), 1583.

Besharati, M., Palangi, V., Moaddab, M., Nemati, Z., Pliego, A. B., & Salem, A. Z. (2021). Influence of cinnamon essential oil and monensin on ruminal biogas kinetics of waste pomegranate seeds as a biofriendly agriculture environment. Waste and Biomass Valorization, 12(5), 2333-2342.

- As per your recommendation, I've already removed the old reference and replaced it with the new one; L343 (Palangi et al., 2022), L355 (Besharati et al., 2020) and other references, please see in text.

Round 2

Reviewer 1 Report

Dear authors,
I have evaluated the revised version of the manuscript identified as animals-2119406. I congratulate you. The current version of the manuscript deserves to be published without any other changes, in my opinion. Good luck